# Long lasting anti-IgG chikungunya seropositivity in the Mayotte population will not be enough to prevent future outbreaks: A seroprevalence study, 2019

Giuseppina Ortu[1,2]*, Gilda Grard[3,4], Fanny Parenton[5], Marc Ruello[1], Marie-Claire Paty[1], Guillaume André Durand[4], Youssouf Hassani[5], Henriette De Valk[1], Harold Noël[1], Unono Wa Maore group[¶]

1 Santé Publique France, Saint-Maurice, France, 2 ECDC Fellowship Programme, Field Epidemiology path (EPIET), European Centre for Disease Prevention and Control (ECDC), Santé Publique France, Saint-Maurice, France, 3 National Reference Center for Arboviruses, French Armed Forces Biomedical Research Institute (IRBA), Marseille, France, 4 Unité des Virus Émergents (UVE: Aix-Marseille Univ-IRD 190-Inserm 1207), Marseille, France, 5 Agence Régionale Santé Mayotte Centre Kinga– 90, Mamoudzou, Mayotte

¶ The complete membership of the author group can be found in the Acknowledgments.
* giuseppina_ortu@outlook.com

**Data Availability Statement:** All relevant data are within the paper and its Supporting Information files.

## Abstract

Chikungunya is an arboviral disease causing arthralgia which may develop into a debilitating chronic arthritis. In Mayotte, a French overseas department in the Indian Ocean, a chikungunya outbreak was reported in 2006, affecting a third of the population. We aimed at assessing the chikungunya seroprevalence in this population, after over a decade from that epidemic. A multi-stage cross sectional household-based study exploring socio-demographic factors, and knowledge and attitude towards mosquito-borne disease prevention was carried out in 2019. Blood samples from participants aged 15–69 years were taken for chikungunya IgG serological testing. We analyzed associations between chikungunya serological status and selected factors using Poisson regression models, and estimated weighted and adjusted prevalence ratios (w/a PR). The weighted seroprevalence of chikungunya was 34.75% (n = 2853). Seropositivity for IgG anti-chikungunya virus was found associated with living in Mamoudzou (w/a PR = 1.49, 95%CI: 1.21–1.83) and North (w/a PR = 1.41, 95%CI: 1.08–1.84) sectors, being born in the Comoros islands (w/a PR = 1.30, 95% CI: 1.03–1.61), being a student or unpaid trainee (w/a PR = 1.35, 95%CI: 1.01–1.81), living in precarious housing (w/a PR = 1.30, 95%CI: 1.02–1.67), accessing water streams for bathing (w/a PR = 1.72, 95%CI: 1.1–2.7) and knowing that malaria is a mosquito-borne disease (w/a PR = 1.42, 95%CI: 1.21–1.83). Seropositivity was found inversely associated with high education level (w/a PR = 0.50, 95%CI: 0.29–0.86) and living in households with access to running water and toilets (w/a PR = 0.64, 95%CI: 0.51–0.80) (n = 1438). Our results indicate a long-lasting immunity from chikungunya exposure. However, the current population seroprevalence is not enough to protect from future outbreaks. Individuals naïve to chikungunya and living in precarious socio-economic conditions are likely to be at high risk of infection in future outbreaks. To prevent and prepare for future chikungunya

**Funding:** The author(s) received no specific funding for this work.

**Competing interests:** This research was supported by Santé Publique France. This does not alter our adherence to PLOS ONE policies on sharing data and materials.

epidemics, it is essential to address socio-economic inequalities as a priority, and to strengthen chikungunya surveillance in Mayotte.

## Introduction

Chikungunya virus (CHIKV) is an RNA virus from the *Togaviridae* family transmitted by *Aedes* spp mosquitoes. Chikungunya disease (CHIK) is responsible for acute febrile illness which can be misdiagnosed as dengue, with which it shares the same main vectors. Classical clinical presentation includes headache, nausea, fatigue, muscle pain, rash and joint swelling, but the most relevant is arthralgia that may disappear within two weeks, (may) persist for months or develop into debilitating chronic arthritis [1–3]. Atypical and severe forms of CHIKV infection with neurological implications have been reported in hospitalized patients [4], as well as in newborns subsequent to peripartum mother-to-infant transmission [5]. Immunity develops after infection, and protects the individual from future infections [6]. To date, no licensed vaccines are available, and treatment is only symptomatic.

CHIK outbreaks have been reported mainly in Asia, Africa, and recently, in the Americas and even in Southern Europe, hence becoming a relevant emerging threat in the northern hemisphere [1]. In Africa, where presumably CHIKV originated [7], outbreaks have been reported since 1952, and specifically in the west and central regions of the continent. In 2005–2006, a CHIKV epidemic that started in Kenya spread to the islands of the Indian Ocean, [8–11], among them, Mayotte, a densely populated French overseas department, where 37.2% of the population were affected [12].

After the 2005–2006 massive outbreak, a surveillance system for chikungunya and dengue viruses was set up in 2008 in Mayotte [13], based on the systematic screening of patients with dengue-like syndromes for malaria by rapid diagnostic test [14] and chikungunya, dengue and leptospirosis by PCR. Although a dedicated surveillance system has been in place in the last decade, no autochthonous CHIKV infections have been reported in Mayotte since 2008 [8, 15–17].

Though immunity in those with past infection is long-lasting [6] and maybe lifelong, CHIKV foci of infection or large scale outbreaks are becoming likelier to occur as the proportion of immune people declines [18]. Indeed, in the last decade, Mayotte has seen a fast and dynamic population growth due to high birthrates and to immigration, to a lesser extent [18]. Because of this context of migrations, the level of immunity in the population might have changed.

After the last epidemic in 2006 [12], no further studies to assess seroprevalence were performed. Considering the dynamic changes of the population in the recent years, a seroprevalence study was carried out to assess the level of herd immunity in the population after more than a decade.

Santé Publique France set up in 2018 a general population survey *Unono Wa Maore* ("About the Mahoran") to assess the population health needs and set up appropriate prevention and health promotion campaigns. For this study, blood samples were collected, and made available for a wide range of analyses, providing the opportunity to assess the seroprevalence of various arbovirus infections, among them CHIKV. Alongside biological sample data collection, participants were interviewed to collect socio-demographic information, and information on attitude and practices with regard to the prevention of vector-borne diseases.

Here, we estimated the seroprevalence of past CHIKV infection in the population of Mayotte in 2018–2019. We also identified sociodemographic factors for CHIKV seropositivity, and described the association between attitudes and practices in arboviral disease prevention,

and CHIKV seroprevalence. These results provide an insight into factors that could be considered as relevant to predict, prevent, control and manage future CHIKV outbreaks in Mayotte.

## Materials and methods

### Setting

Mayotte is a densely populated island of 374 km$^2$ with roughly 250,000 people; generally hot and humid, it is characterized by high rainfall, especially in the center-west and north parts of the island. Currently, roughly half of the Mahoran population holds a foreign nationality and is below age 18 [18]. Foreign population, especially coming from the Comoros islands, and high population growth rate have been particularly evident in the north-eastern part of the Island. Although the Mahoran population is in rapid growth, the number of young adults, especially between age 15 and 24, is decreasing due to the migratory outward flow of this generation towards metropolitan France [19].

The majority of the Mahoran population is poor and lives in precarious conditions, e.g. in inadequate housing; this situation has been particularly worrying for immigrants coming from neighboring islands [20, 21]. In 2017, roughly 39% of the population was reported to live in metal sheet, wood, vegetation or mud housing, and without access to running water or toilets inside their house [19, 20]. The population with these precarious living conditions is especially concentrated in Mamoudzou and surrounding communes [22], and some villages in the north [23].

### Study design

The Unono wa Maore study, conducted between December 2018 and June 2019, consists of a cross-sectional household-based study exploring health status, social determinants of health, vaccination coverage in children, behaviors and attitudes toward sex, prevalence of major chronic diseases and sexually transmitted diseases, childcare and mosquito-borne disease prevention.

We used a multi-stage cluster sampling design to select study participants. First, we randomly selected addresses of buildings from the 2017 register of localized buildings (RIL), which contains all the housing addresses necessary for the population census. We stratified addresses by number of households: (i) one household, (ii) 2 to 9 households, (iii) 10 or more. When there were up to 10 households per address, all households were included. Otherwise, a maximum of 10 households were randomly selected. In each household, we randomly selected a maximum of five individuals of which 3 individuals were aged 15 years and older. Further details on the methodology can be found elsewhere [24].

### Data collection

Trained interviewers conducted face-to-face questionnaire interviews in French and two local languages, Kibushi and Shimaore.

Two questionnaires were administered: one comprehensive questionnaire for the randomly selected first participant aged 15 and older included in the household, and a short one for the other participants of the same household. Both questionnaires explored participants' socio-demographic characteristics; alcohol, tobacco, and drugs use; perceived health; hygienic practices; vaccination; perception and behavior related to vector borne diseases. Only the comprehensive questionnaire collected data on family origin, socio-economic status and education, healthcare seeking behavior; food intake, and sexual behavior. Additionally, interviewers collected information on the type of dwelling as well as access to water.

Following the interviews, participants made an appointment with a nurse for anthropometric measurements and collection of biological samples. Only participants over 15 years old were asked to provide biological samples, and only those aged 15–69 years were tested for past CHIK infection.

## Laboratory analysis

Blood Samples of 5 mL collected by venipuncture were frozen at –20˚C and subsequently defrosted for CHIKV total IgG antibodies testing using an in-house direct ELISA on inactivated viruses (in-house prepared antigens) [25]. The threshold for ELISA positivity was set at a ratio of measured optical density to negative antigen >3.

The National Reference Center for arboviral diseases in Marseille (Institut de Recherche biomédicale des Armées) conducted all serologic testing.

## Statistical analysis

Statistical analyses were performed with STATA 16.0 (Statacorp. College Station, TX, USA) using the "svy" command to account for the study design. Age, gender, and location-adjusted weights were calculated and used to estimate IgG weighted Prevalence Ratios (wPR), and their 95% confidence intervals for each characteristic (sociodemographic, knowledge, attitude to personal protection and vector control, and health status). We opted to report the weighted seroprevalence by sectors, a group of communes based on an administrative division already in place, with socio-demographic and logistic similarities, and existing road links. In S1 File. Map of Mayotte with communes and corresponding sectors on the right table, we listed the communes grouped by sectors.

We estimated anti CHIKV IgG seroprevalence among all individuals aged 15 to 69 that provided a blood sample. For the descriptive analysis and bi- and multivariable analysis, we used the data obtained from all participants who answered to the comprehensive questionnaire.

For the univariate analysis (binomial regression), we used anti CHIKV IgG seropositivity as binary outcome and calculated weighted Prevalence Ratios (svy-weighted analysis).

We tested independent associations between serological status as binary outcome and explanatory variables using a Poisson model with robust variance, and computed weighted and adjusted prevalence ratios (w/a PR). We performed three different Poisson models exploring (i) all variables selected, (ii) sociodemographic factors only, (iii) and environmental/housing and knowledge and attitude on prevention and control of arboviral diseases. In these models, we included explanatory variables that were associated with seropositivity in the univariable analysis with a P-value ≤0.20. Collinearity was checked via variance inflation factor (VIF) and Spearman's rank correlation test. Although the VIF test provided an average value of 1.62 (therefore not suggesting collinearity), a few variables appeared to have slightly higher VIF values (around 2–3); therefore, we opted to run the Spearman's test. For the latter, when two variables had a pairwise correlation coefficient >0.5 [26], we combined them and reduced the number of categories to reach a final list of variables to be used in the three multivariable models (see also S6 File. Information on datasets, weights and variables used in the analysis, for further clarifications). In each of these models, a backward stepwise elimination [27] and retention procedure of covariables with a *P* value< 0.05 with a svy-adjusted Wald test was used to finalize each model. Interactions among variables were also explored.

## Ethics statement

The UNONO WA MAORE study is interventional research involving the human being with minimal risks and constraints and with a public interest purpose (L.1121-1 2˚ of the public

health code). Therefore, the methodology of the study complies with the provisions of the public health code relating to this type of research and in accordance with the Declaration of Helsinki.

The protocol was validated by the French ethical committee for biomedical research, the Committee for the Protection of Persons (CPP, no. 2017-A02782-51), and complied with MR001 reference methodology (agreement from the National Commission for Informatics and Freedoms of 25 September 2018, no. 918233.

Information on the study and consent forms were read to the participants and a written informed consent was obtained from them or from a legal representative when participants were ≤17 years old. Samples and data were anonymized at the time of collection; therefore, sample testing and data analysis were conducted anonymously [24].

## Results

### Household and population characteristics

The survey was performed in all communes of the island. 2716 households were included in the survey, and 4817 individuals between the age of 15 and 69 accepted to participate, of whom 2853 were tested for past CHIKV infection. Among those tested for past CHIKV infection, 2778 answered the questionnaires among whom 1438 answered the comprehensive questionnaire.

Participants tested for CHIKV (n = 2778) had a median age of 34 years (min 15, max 69) and were mainly between age 30 and 49 (43.5%), and female (63.8%). Most of them resided in the Mamoudzou sector (36.5%) (Table 1).

Among those who answered the comprehensive questionnaire (n = 1438), the median age was 35 years, (min 15, max 69), 38.4% (n = 552) were born in Mayotte and 52.5% (n = 754) in the Comoros Islands. 63.6% (n = 913) did not receive any type of education, 74.7% (n = 1072) were unemployed, or pensioners or students, 38.6% (n = 555) did not have any social security and 41.0% (n = 589) lived in a household with ≥4 children (Table 1). Housing was 40.4% (n = 581) traditional wooden houses (called "bangas") or wood-sheet metal hut, 49.5% (n = 712) did not have running water and 51.7% (n = 744) did not have toilets inside the

**Table 1. Weighed seroprevalence of chikungunya by age, sex and residence sector.**

| Characteristics | | Total | | CHIK positive N | CHIK positive (%) | IgG$_{CHIK}$ (+) Weighed prevalence (%) | 95% CI | |
|---|---|---|---|---|---|---|---|---|
| **Tested for arboviral exposure** | | 2853 | | **1007** | 35.30 | **34.75** | 32.32 | 37.26 |
| **N tested and interviewed*** | 2778 | **N** | **%** | **CHIK positive n** | **CHIK positive (%)** | **IgG$_{CHIK}$ (+) Weighed prevalence (%)** | **95% CI** | |
| **Age (Years)** | 15–17 | 353 | 12.71 | 112 | 31.73 | 31.43 | 25.78 | 37.69 |
| | 18–29 | 716 | 25.77 | 257 | 35.89 | 33.55 | 29.32 | 38.06 |
| | 30–49 | 1207 | 43.45 | 443 | 36.70 | 36.12 | 32.62 | 39.78 |
| | 50–69 | 502 | 18.07 | 165 | 32.87 | 36.02 | 31.08 | 41.26 |
| **Sex** | Male | 1007 | 36.25 | 365 | 36.25 | 36.26 | 32.63 | 40.06 |
| | Female | 1771 | 63.75 | 612 | 34.56 | 33.45 | 30.59 | 36.45 |
| **Sector of residence** | Centre | 773 | 27.83 | 252 | 32.60 | 31.59 | 27.72 | 35.73 |
| | Mamoudzou | 1015 | 36.54 | 439 | 43.25 | 41.78 | 37.83 | 45.83 |
| | North | 255 | 9.18 | 103 | 40.39 | 40.85 | 33.63 | 48.5 |
| | Petite-Terre | 442 | 15.91 | 126 | 28.51 | 29.93 | 25.03 | 35.33 |
| | South | 293 | 10.55 | 57 | 19.45 | 17.19 | 12.44 | 23.29 |

*Short questionnaire only

premises. Additional information is available in **S2 File**. Weighed chikungunya seroprevalence, weighed prevalence ratios and univariate analysis for all covariates.

## Overall anti-CHIKV IgG seroprevalence and seroprevalence by age, sex and sector

IgG positive to CHIKV were found in 35.3% (n = 1007) of those tested (n = 2853). The overall weighted seroprevalence of anti-CHIKV IgG in Mayotte was 34.75% (95%IC [32.3–37.3], n = 2853) (Table 1). In those tested and having completed a questionnaire, there were no significant differences in seroprevalence between the different age groups nor by sex. Mamoudzou and North were the sectors with the highest (weighted) seroprevalence (41.8%, 95%CI [37.8–45.8], and 40.9%, 95%CI [33.6–48.5], respectively), and South the one with the lowest (17.2%, 95%CI [12.4–23.3]) (Fig 1).

## Determinants of anti-CHIKV IgG seropositivity

For the analysis of *anti-CHIKV IgG* seropositivity determinants (n = 1438), results from the univariable analysis can be found in S2 File. Weighed chikungunya seroprevalence, weighed prevalence ratios and univariate analysis for all covariates. Among those variables that were

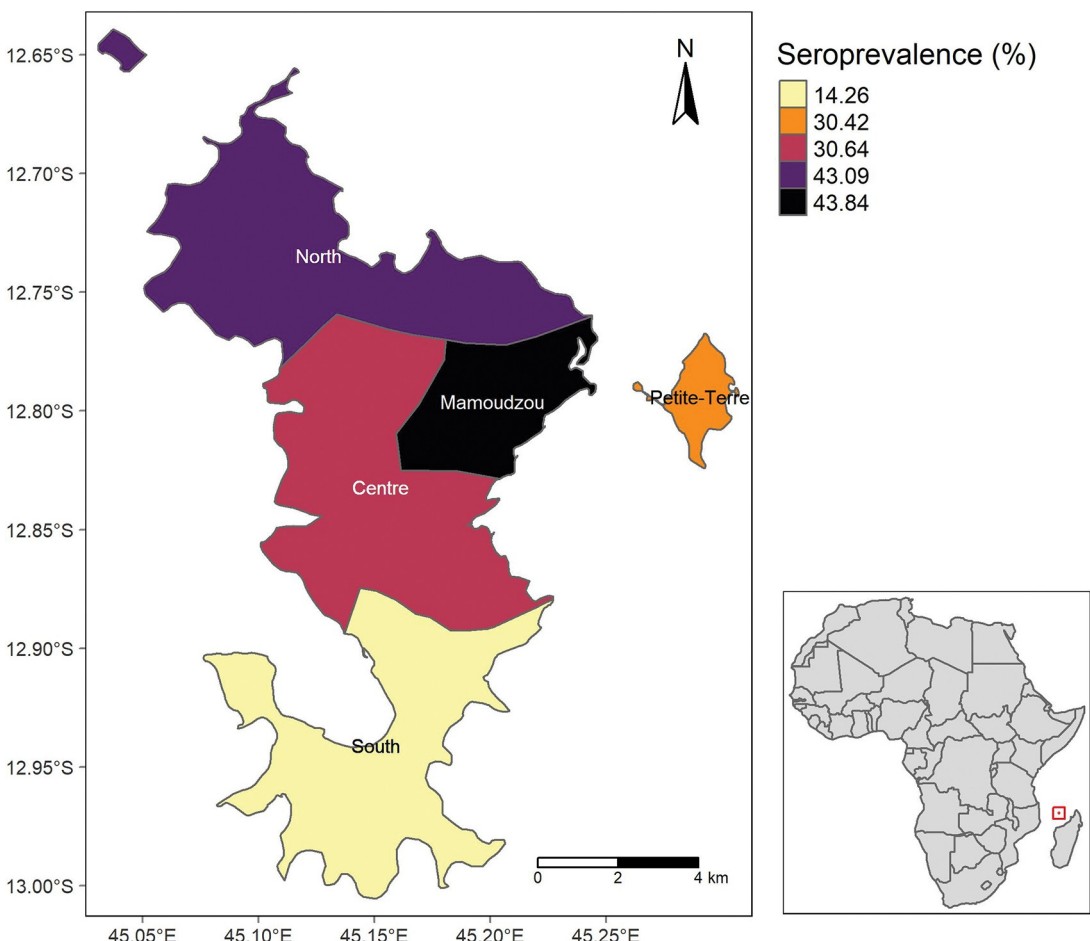

**Fig 1. CHIKV seroprevalence in North, Mamoudzou, Petite-Terre, Center and South sectors of Mayotte.**

significantly associated with past CHIKV exposure, availability of social security or social security plus complementary health insurance were negatively associated with seropositivity. On the contrary, accessing the river for daily washing, being very annoyed by mosquitoes, covering water reservoirs, and smoking habits were all factors associated with anti-CHIKV IgG seropositivity. All these variables were initially added to the multivariable models as found statistically significant in the univariable analysis; however, some of them were subsequently eliminated during the backwards stepwise process as no longer significant in the models. More information on other variables that were tested in the univariable analysis can be found in S2 File. Weighed chikungunya seroprevalence, weighed prevalence ratios and univariate analysis for all covariates.

We performed three models, the first one in which we included all the covariates that from the Spearman test did not appear to create collinearity (M1), and two other models in which we considered only those variables related to specific domains (sociodemographic determinants–M2), and attitude and knowledge related to arboviral disease prevention and vector control and environmental/housing factors (M3) (Table 2).

Looking at the model with all the covariates (M1), residing either in Mamoudzou (w/a PR = 1.49, [95%CI: 1.21–1.83]) or North (w/a PR = 1.41 [1.08–1.84]) was associated with higher CHIKV seroprevalence compared to residing in Centre or Petite Terre. Living in the southern sector was associated with the lowest level of CHIKV seroprevalence (w/a PR = 0.49 [0.32–0.77]). Participants originating from Comoros Islands were more likely to be anti-CHIKV IgG positive than those born in Mayotte (w/a PR = 1.30 [1.03–1.61]).

Compared to individuals with no education, persons with university degrees were less likely to have anti-CHIKV IgG (w/a PR = 0.49 [0.29–0.85]). Finally, individuals having running water and a toilet in the household were less likely to have been infected with CHIKV (w/a PR = 0.64 [0.51–0.80]).

M2 exploring socio-demographic covariates/factors, showed that compared to individuals with employment, anti-CHIKV IgG seroprevalence was significantly higher in students (w/a PR = 1.35[1.01–1.81]). In M3 exploring attitudes on mosquito borne diseases prevention and environmental factors, anti-CHIKV IgG seroprevalence was higher among individuals living in traditional wooden houses or wood-sheet metal huts compared to individuals living in houses on cement or apartment (w/a PR = 1.31 [1.01–1.68]). M3 results corroborated the association of high anti-CHIKV IgG seroprevalence with not having running water or a toilet in the household (w/a PR = 0.61 [0.47–0.81]), compared to those individuals with both, as found in M1. M3 model also indicated that going to the river for bathing routinely (w/a PR = 1.72 [1.10–2.7]), and knowing that malaria is transmitted by mosquito bites (w/a PR = 1.42 [1.03–1.97]) were associated with anti-CHIKV IgG seroprevalence.

## Discussion

We observed an overall weighed seroprevalence of anti-CHIKV IgG of 35% (95%CI: 32–37) amongst people aged 15 and above. This estimate is very similar to the seroprevalence of anti-CHIKV IgG as found in 2006 just after the outbreak in Mayotte, suggesting that there was no or minimal transmission of the virus in the interim [12]. CHIKV re-emerged in the Indian Ocean in 2009–2010 [28], namely in Reunion island and Madagascar, but not in Mayotte. Since 2006, no further cases of CHIKV infection have been reported in Mayotte [15–17, 29], and our findings therefore suggest long-lasting seropositivity against CHIKV in the Mayotte population.

According to our results, 65% of the Mayotte population aged 15 to 69 remains susceptible to CHIKV infection. Although we did not sample children below 15 years, this age group has

**Table 2. Weighed and adjusted prevalence ratios for CHIKV seropositivity (w/a PR) according to sociodemographic, environmental and attitude factors.** Model 1 includes all covariates, model 2 sociodemographic variables, and model 3 variables related to knowledge /attitudes in arboviral prevention and control and environment/ housing factors.

| Characteristics | | Model 1 (all covariates) (n = 1435) | | | Model 2 (n = 1432) | | | Model 3 (n = 1437) | | |
|---|---|---|---|---|---|---|---|---|---|---|
| | | w/a PR (lincom) | 95% CI | P value | w/a PR | 95% CI | P value | w/a PR | 95% CI | P value |
| **Sociodemographic factors** | | | | | | | | | | |
| Age (Years) | 15–17 | ref | | | ref | | | ref | | |
| | 18–29 | 1.04 | 0.74 1.45 | 0.834 | 1.16 | 0.76 1.78 | 0.48 | 1.01 | 0.73 1.41 | 0.94 |
| | 30–49 | 0.93 | 0.67 1.3 | 0.673 | 1.07 | 0.68 1.7 | 0.762 | 0.95 | 0.69 1.3 | 0.73 |
| | 50–69 | 0.99 | 0.69 1.43 | 0.94 | 1.12 | 0.69 1.83 | 0.638 | 1.01 | 0.7 1.5 | 0.934 |
| Sex | Male | ref | | | ref | | | ref | | |
| | Female | 0.94 | 0.79 1.11 | 0.448 | 0.92 | 0.76 1.1 | 0.362 | 1.01 | 0.84 1.2 | 0.939 |
| Sector of residence | Centre | ref | | | ref | | | | | |
| | Mamoudzou | 1.49 | 1.21 1.83 | $<10^{-4}$ | 1.44 | 1.16 1.78 | **0.001** | | | |
| | North | 1.41 | 1.08 1.84 | **0.011** | 1.45 | 1.11 1.9 | **0.007** | | | |
| | Petite-Terre | 1.05 | 0.78 1.4 | 0.772 | 1.01 | 0.75 1.37 | 0.842 | | | |
| | South | 0.50 | 0.32 0.77 | **0.002** | 0.48 | 0.31 0.75 | **0.001** | | | |
| Education level | No education | ref | | | ref | | | | | |
| | Education below university degree | 1.01 | 0.82 1.24 | 0.911 | 0.94 | 0.75 1.2 | 0.599 | | | |
| | University degree / higher education | 0.50 | 0.29 0.86 | **0.012** | 0.44 | 0.25 0.77 | **0.004** | | | |
| | Other education | 0.66 | 0.33 1.32 | 0.238 | 0.64 | 0.32 1.3 | 0.208 | | | |
| Professional situation | Employed | | | | ref | | | | | |
| | Unemployed | | | | 1.45 | 0.96 2.25 | 0.074 | | | |
| | Student, unpaid trainee / intern | | | | 1.35 | 1.01 1.81 | **0.041** | | | |
| | Inactive (pensioner, home maker etc,) | | | | 1.1 | 0.84 1.47 | 0.469 | | | |
| | Work placement, training course, apprenticeship, or paid internship | | | | 1.46 | 0.89 2.4 | 0.133 | | | |
| Place of birth | Mayotte | ref | | | ref | | | | | |
| | Comoros | 1.3 | 1.03 1.61 | **0.026** | 1.43 | 1.15 1.78 | **0.001** | | | |
| | Madagascar | 1.08 | 0.7 1.68 | 0.728 | 1.04 | 0.64 1.68 | 0.885 | | | |
| | Metropolitan France and other countries | 0.54 | 0.22 1.32 | 0.174 | 0.47 | 0.19 1.15 | 0.097 | | | |
| **Household & environment** | | | | | | | | | | |
| Habitation type | House on cement | | | | | | | ref | | |
| | Traditional wooden house or wood-sheet metal hut | | | | | | | 1.30 | 1.02 1.67 | **0.035** |
| | Apartment | | | | | | | 1.23 | 0.82 1.85 | 0.312 |
| Interaction term: | Habitation type and access to the river for washing dishes or clothes | | | | | | | no significant | | |
| Access to the river for washing clothes or dishes | Never | | | | | | | ref | | |
| | Sometimes/often for either reason | | | | | | | 1.17 | 0.65 2.1 | 0.593 |
| | Always | | | | | | | 0.53 | 0.12 2.26 | 0.391 |
| Running water and WC in the household | No | ref | | | | | | ref | | |
| | Yes | 0.64 | 0.51 0.80 | $<10^{-4}$ | | | | 0.61 | 0.47 0.79 | $<10^{-4}$ |
| Go to the river for bathing | Never | | | | | | | ref | | |
| | Sometimes/often | | | | | | | 1.02 | 0.66 1.59 | 0.911 |
| | Always | | | | | | | 1.72 | 1.1 2.7 | **0.018** |
| **Attitudes to mosquito-borne disease prevention** | | | | | | | | | | |

*(Continued)*

**Table 2.** (Continued)

| Characteristics | | Model 1 (all covariates) (n = 1435) | | | Model 2 (n = 1432) | | | Model 3 (n = 1437) | | | |
|---|---|---|---|---|---|---|---|---|---|---|---|
| | | w/a PR (lincom) | 95% CI | P value | w/a PR | 95% CI | P value | w/a PR | 95% CI | | P value |
| Understanding whether transmission of malaria is via mosquitoes | No | | | | | | | ref | | | |
| | Yes | | | | | | | 1.42 | 1.03 | 1.97 | **0.033** |
| | no answer | | | | | | | 1.30 | 0.9 | 1.9 | 0.162 |

most likely not been substantially exposed to CHIKV in Mayotte since the last major outbreak in 2005. Pooling together the 44% of the population under 15 years [19] and the estimated number of susceptible individuals, roughly 80% of the Mahoran population remained is susceptible to CHIKV.

It appears, inevitable that new outbreaks could take place ifit is therefore, inevitable the occurrence of future epidemics in Mayotte if CHIK is reintroduced in this population. For this reason, particular attention to CHIKV transmission is warranted to anticipate and prepare for a potential CHIKV introduction and outbreaks in Mayotte.

We studied determinants of CHIKV seropositivity among which sector of residence was an important risk factor for past CHIKV infection. In agreement with a serosurvey performed after the 2006 CHIKV outbreak [30], Mamoudzou and the Northern sector of Mayotte had higher seroprevalence than the center and the mid-west areas (Centre sector). Seroprevalence was lowest in the Southern sector. This geographical distribution of cases may reflect higher past CHIKV transmission in areas with high annual rainfall (northern part of the island), densely populated (northern and north-eastern areas), or urban areas such as Mamaoudzou, Koungou, Dzouadzi and Pamandi.

Geographical area of birth was also significantly associated with anti-CHIKV IgG seroprevalence, with individuals born in the Comoros having higher seroprevalence compared to those born in other geographical areas. This result is in agreement with the higher seroprevalence reported during the CHIKV infection outbreak in these islands in 2005–2006 (over 60%), compared to the seroprevalence reported in Mayotte (around 37%) [9, 30, 31].

Knowledge of other mosquito-borne diseases such as malaria, appeared to be positively associated with the presence of CHIKV antibodies. This association could reflect a heightened awareness of mosquito-borne infectious diseases among individuals affected by malaria or chikungunya.

Other factors associated with the presence of anti-CHIKV IgG were lower education and lack of access to running water or toilets in the house. Models 2 and 3 suggest that living in a traditional wooden house or wood-sheet metal hut, being a student (and unemployed) and always using the river for bathing are all associated with the presence of anti CHIKV antibodies. These results are in agreement with the data obtained in the seroprevalence survey performed in 2006 [30], where factors indicative of socio-economic disadvantage (e.g., length of schooling, makeshift housing) were found associated with anti-CHIKV IgG seropositivity.

Altogether, these results suggest that today's socio-economically disadvantaged groups that were not exposed to CHIKV infection in 2006, may be a high risk of infection in future CHIKV outbreaks, compared to other groups.

Improvement of housing conditions should be part of CHIK infection prevention, and more broadly, part of prevention strategies for all arboviral diseases. Makeshift and precarious housing (where generally, mosquitoes breeding sites are found more frequently) associated with high population density, increase substantially the chance of transmission of endemic

arboviral diseases, making this type of environment prone to disease outbreaks. In Mayotte, several communes in the Mamoudzou sector have the highest number of people living in slums, under precarious socio-economic conditions [32]. Arboviral diseases preparedness plans could target these areas as a priority, as well as people living in similar conditions in the north part of the island, and improvement of housing and environmental conditions should be part of these plans.

## Limitations

Our study has some limitations. Data on individuals younger than age 15 and older than 69 were not available, feasibility of sample collection and ethical issues being the main reasons for not including them; hence, the results from this cross sectional seroprevalence study cannot be extrapolated to these age groups. Not including individuals above age 69 ($\sim$<1% of the population), has likely not impacted our results as based on data from the last outbreak [12], CHIKV seroprevalence in the population between age 56 and 79 was very similar to other age groups ($\sim$ between 23 and 50%). Not including children below age 15, has eliminated a part of the population which was most likely not exposed to CHIKV during the outbreak in 2006. With the addition of this young (naïve) population, the estimated CHIKV seroprevalence is likely to be lower than our estimates.

Although we could estimate CHIKV seroprevalence differences among the different sectors with statistical significance, the study was not designed to determine risk factors at the sector level but only at the department level. For this reason, we could not estimate the association between seroprevalence and the selected determinants in sectors, which could have been more informative and useful from a public health response perspective.

## Conclusions

This study has indicated the presence of antibodies against CHIKV in a third of the Mahoran population between the age 15 and 69. This finding implies that two thirds of the adult population is susceptible to CHIKV infection, namely those living under better socio-economic conditions, and children below the age of 15. Our study indicates also that socio-economically disadvantaged groups naïve to CHIKV infection are at higher risk than others in future epidemics.

In this context where the majority of the population is still naïve with regard to CHIKV, and the risk of another epidemic is high due to the continuous transmission of chikungunya virus in other parts of the world, a vaccine would be extremely effective in reducing the risk of infection and future outbreaks. Unfortunately, CHIKV vaccines are still in a development phase and yet to be deployed [33].

Despite the worldwide transmission of chikungunya throughout the tropics, no vaccines are available at the moment. Although planning for chikungunya vaccine trials might be challenging, the global effort to develop and distribute vaccines should be accelerated as core preparedness and response to upcoming CHIK outbreaks [34]. Serosurveys should be conducted whenever possible at least after each CHIK outbreak to monitor its spread and assess population immunity and inform efficient vaccine rollout.

In the absence of an effective vaccine, a viable strategy for preventing CHIKV infections (and other arboviral diseases) is to roll back social inequalities by improving the living conditions in rural, and densely populated urban areas, especially in the center and north part of the island.

Maintaining the screening for CHIKV and other pathogens in febrile patients as it is currently done in Mayotte, is paramount to prepare for future epidemics, as well as insuring a

close monitoring of CHIKV circulation in the Indian Ocean region and neighboring African countries. These routine assessments would help in understanding whether CHIKV is circulating, and therefore, informing on whether public health measures should be initiated to prevent further spreading.

Finally, it is of utmost importance to continue to raise awareness among the whole population of Mayotte on how to prevent arboviral infections.

## Supporting information

**S1 File. Map of Mayotte with communes and corresponding sectors in the table on the right.**
(DOCX)

**S2 File. Weighed chikungunya seroprevalence and weighed prevalence ratios for all covariates (N = 1438, it may slightly differ for some of the covariates due to missing values).**
(DOCX)

**S3 File. Dataset CHIK analysis_PLOSOne dataset_n2853.**
(DTA)

**S4 File. Dataset CHIK analysis_PLOSOne dataset_n2778.**
(DTA)

**S5 File. Dataset CHIK analysis_PLOSOne dataset_n1438.**
(DTA)

**S6 File. Information on datasets, weights and variables used in the analysis.**
(DOCX)

## Acknowledgments

The authors are especially grateful to the investigators of the Unono Wa Maore group: Marc Ruello, Marion Fleury, Jean-Baptiste Richard, Jean-Louis Solet, Laurent Filleul, Delphine Jezewski-Serra, Julie Chesneau

## Author Contributions

**Conceptualization:** Giuseppina Ortu, Gilda Grard, Fanny Parenton, Marc Ruello, Marie-Claire Paty, Guillaume André Durand, Youssouf Hassani, Henriette De Valk, Harold Noël.

**Data curation:** Giuseppina Ortu, Marc Ruello, Harold Noël.

**Formal analysis:** Giuseppina Ortu, Gilda Grard.

**Funding acquisition:** Henriette De Valk, Harold Noël.

**Investigation:** Gilda Grard, Fanny Parenton, Marc Ruello, Guillaume André Durand, Youssouf Hassani.

**Methodology:** Giuseppina Ortu, Gilda Grard, Fanny Parenton, Marc Ruello, Marie-Claire Paty, Guillaume André Durand, Youssouf Hassani, Henriette De Valk, Harold Noël.

**Project administration:** Marc Ruello, Marie-Claire Paty, Youssouf Hassani, Harold Noël.

**Resources:** Gilda Grard, Fanny Parenton, Marc Ruello, Marie-Claire Paty, Guillaume André Durand, Youssouf Hassani, Henriette De Valk, Harold Noël.

**Software:** Giuseppina Ortu.

**Supervision:** Gilda Grard, Guillaume André Durand, Henriette De Valk, Harold Noël.

**Validation:** Giuseppina Ortu, Gilda Grard, Fanny Parenton, Marc Ruello, Marie-Claire Paty, Guillaume André Durand, Henriette De Valk, Harold Noël.

**Visualization:** Giuseppina Ortu.

**Writing – original draft:** Giuseppina Ortu.

**Writing – review & editing:** Giuseppina Ortu, Gilda Grard, Fanny Parenton, Marc Ruello, Marie-Claire Paty, Guillaume André Durand, Youssouf Hassani, Henriette De Valk, Harold Noël.

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
