## [Decision Letter · Decision Letter 0]

3 Feb 2023

PONE-D-22-35076Chikungunya long lasting immunity in the Mayotte population will not be enough to prevent future outbreaks: A seroprevalence study, 2019.PLOS ONE

Dear Dr. Ortu,

Thank you for submitting your manuscript to PLOS ONE. After careful consideration, we feel that it has merit but does not fully meet PLOS ONE’s publication criteria as it currently stands. Therefore, we invite you to submit a revised version of the manuscript that addresses the points raised during the review process.

Two of the three reviewers agree that your manuscript should require a major revision. Please address all of their concerns before resubmitting.

We look forward to receiving your revised manuscript.

Kind regards,

Joel Mossong, PhD

Academic Editor

PLOS ONE

Journal Requirements:

"This research was supported by Santé Publique France"

Reviewers' comments:

Reviewer's Responses to Questions

**Comments to the Author**

1. Is the manuscript technically sound, and do the data support the conclusions?

Reviewer #1: Partly

Reviewer #2: Yes

Reviewer #3: Partly

2. Has the statistical analysis been performed appropriately and rigorously? 

Reviewer #1: No

Reviewer #2: Yes

Reviewer #3: Yes

3. Have the authors made all data underlying the findings in their manuscript fully available?

Reviewer #1: No

Reviewer #2: Yes

Reviewer #3: Yes

4. Is the manuscript presented in an intelligible fashion and written in standard English?

Reviewer #1: Yes

Reviewer #2: Yes

Reviewer #3: Yes

5. Review Comments to the Author

Reviewer #1: The study investigated the susceptibility level of CHIKV in human population in Mayotte using the serological data randomly sampled in the region. They concluded that the immunity in CHIKV is long-lasting and the current immunity from the previous outbreak is not enough to protect from future outbreaks. I found this conclusion a bit inaccurate and may need to revise some discussion points. Some of the analyses in the manuscript should also be revised.

The previous outbreak in Mayotte was in 2005-2006 and the serological data from 2019 was investigated in this study. Unfortunately, the serological data is missing age group < 15, which contains crucial information to know if there are any additional outbreaks in the region 15 years since 2005. The authors were aware of this problem and already discussed it. Besides no report of CHIKV transmission in the region since 2005, by observing that the seroprevalence in the population remains the same compared to the previous study across all age groups, we have a stronger evidence that there were no activity of the virus since 2005. This point should be added to the discussion section.

I suggest the authors should assess for multicollinearity using VIF rather than just collinearity. I also wonder why the threshold of 0.5 was chosen for the pairwise correlation to reduce the number of variables. Why not 0.7 or any other numbers? I suggest the authors attempt to reduce the dimension of the variables by other means such as stepwise regresion.

The results from the univariate analysis should be summarized or discussed somewhere in the manuscript.

I also wonder why the authors perform the analyses on the three models with different fixed set of covariates. Some kind of variables selection procedure here may make the method more robust.

“Based on an estimated basic reproduction number of 2.8 reported for areas where Aedes albopticus is the primary vector (29), roughly 65-70% of the population should be immune to CHIKV to avoid an epidemic; therefore, the possibility of future epidemics in Mayotte cannot be excluded.” I found this point to be problematic. The 65-70% figure is derived from the R0 of 2.8 assuming that the population is homogenous. This in reality is not true for aborviruses, where their transmission dynamic are highly heterogenous. Given the heteogenousity, the herd immunity threshold can be much lower than 65-70%, hence current immunity level in the population maybe enough to prevent epidemic to happen in the near future. Furthermore, the authors seems to ignore the uncertainty of the R0 estimation, which lead to the uncertainty in the herd immunity threshold as well. I also found the conclusion for this point is weak, since in the future, as the susceptible pool is increasing over time from the newborns, it is inevitable that “the possibility of future epidemics in Mayotte cannot be excluded.”

“Knowledge of other mosquito-borne diseases such as malaria, appeared to be positively associated with the presence of CHIKV antibodies. This association could reflect a heightened awareness of mosquito-borne infectious diseases among individuals affected by malaria or chikungunya.” I failed to see the authors’ logic of this point. I believe there is a potential confounder here, knowledge of malaria and presence of CHIKV antibodies are both correlated with high risk of arboviruses infection.

The lines in the manuscript should be numbered.

Reviewer #2: Dear authors,

Excellent work conducting this work and conveying key information about a neglected disease. Please see some specific comments below:

Is it possible to add a map illustrating the different regions of Mayotte and their respective seroprevalence? A figure would be helpful for readers unfamiliar with the island when discussing the geographic findings of the study.

Introduction, 2nd paragraph, line 4, remove “has” in “where presumably CHIKV has originated”

Introduction, final paragraph, line 4, change “insight of” to “insight into”

Setting, paragraph 1, line 3, change “is holding” to “holds”

Study design, paragraph 1, line 1, change “consists in” to “consists of”

Data collection, paragraph 2, line 7, change “Besides,” to “Additionally,”

Determinants of anti-CHIKV IgG seropositivity, paragraph 3, line 1, change “being resident either in Mamoudzou” to “residing in either Mamoudzou”

Determinants of anti-CHIKV IgG seropositivity, paragraph 3, line 2, remove “the” in “associated with the higher”

Determinants of anti-CHIKV IgG seropositivity, paragraph 3, line 3, change “being resident” to “residing”

Determinants of anti-CHIKV IgG seropositivity, paragraph 4, line 3, change “toilette” to “toilet”

Discussion, paragraph 2 - “therefore we should add this population to the group of susceptible individuals.” Is it possible to provide a quantitative estimate of this value? Ie. if you assume all members of the population below 15 are seronegative and calculate a new percentage including these individuals, by how much does the 65% susceptibility estimate increase?

Reviewer #3: Ortu et al. performed a serosurvey and questionnaire in Mayotte, where a CHIKV outbreak took place in 2006. The authors performed IgG ELISA to determine previous CHIKV exposure and applied questionnaires exploring socio-demographic factors and knowledge of mosquito-borne diseases. Although the part of the study looking at CHIKV IgG could be substantially expanded upon, by doing quantitative ELISA, neutralization assays, and T-cell assays to assess the population immunity, the igG data presented here will be a valuable addition to the literature for understanding CHIKV present and future outbreaks.

Major comments:

1.The title of the manuscript claims that the current immunity will not be enough to prevent future outbreaks. This claim is based on a report that estimates that at least 65% of the population would need to be immune to prevent future epidemics. However, in this paper the authors only perform a positive/negative IgG, and do not assess other parts of the immune system that could give partial immunity to CHIKV such as T-cells, and proportion of neutralizing antibodies, cross-neutralizing antibodies from similar alphaviruses. Therefore, the authors should consider changing the title to reflect their data accurately.

2. The authors stored the sera at -20C. Please add how long the sera was stored for. How long are IgGs stable for at -20C?

3. The references need to be improved on several occasions:

•The authors cite a manuscript in preparation; this is highly unusual, and I strongly suggest removing that citation. The details on the methodology should be included in this manuscript, in the methods or supplemental files, or a published source could be cited.

•References 15-17 look like they need more information, is that citing a book chapter?

•While citing reputable websites is acceptable, the authors cite many websites. I would suggest reducing the number of website citations wherever possible and adding the date accessed for any that will be kept. Links to websites tend to break with time and no longer be accessible.

•Reference 31 is all in caps, while the others are not.

4. More details are needed regarding the ELISA. Please clarify if the ELISA is for total IgG, IgG-1 or other. Please specify how the ELISA threshold was chosen. If it was recommended by a manufacturer or based on previous experience or statistical method.

5. Since the authors are working with human blood from a region that may have CHIKV present, the authors should specify how the samples were handled, ie/ level of containment, any heat inactivation done, if anything was done in case blood was collected from a patient with CHIKV symptoms. Although the samples were not known to contain CHIKV, this virus is typically BSL-3 in Europe and the US.

Minor comments:

1. Would it be possible for the authors to use a known sample as standard curve to quantify the IgGs?

2. The "attitudes to mosquito-borne disease prevention" data is not showing on table 2, I think it was cut off.

6. PLOS authors have the option to publish the peer review history of their article (what does this mean?). If published, this will include your full peer review and any attached files.

Reviewer #1: No

Reviewer #2: No

Reviewer #3: No

---

## [Author Response · Author response to Decision Letter 0]

2 Apr 2023

Manuscript 

PONE-D-22-35076

Chikungunya long lasting immunity in the Mayotte population will not be enough to prevent future outbreaks: A seroprevalence study, 2019.

PLOS ONE

REPLY TO REVIEWERS

Reviewer #1: 

The study investigated the susceptibility level of CHIKV in human population in Mayotte using the serological data randomly sampled in the region. They concluded that the immunity in CHIKV is long-lasting and the current immunity from the previous outbreak is not enough to protect from future outbreaks. I found this conclusion a bit inaccurate and may need to revise some discussion points. Some of the analyses in the manuscript should also be revised.

The previous outbreak in Mayotte was in 2005-2006 and the serological data from 2019 was investigated in this study. Unfortunately, the serological data is missing age group < 15, which contains crucial information to know if there are any additional outbreaks in the region 15 years since 2005. The authors were aware of this problem and already discussed it. Besides no report of CHIKV transmission in the region since 2005, by observing that the seroprevalence in the population remains the same compared to the previous study across all age groups, we have stronger evidence that there were no activity of the virus since 2005. This point should be added to the discussion section.

Answer:

The authors acknowledge your general feedback – thank you. We mentioned the evidence of not circulation of CHIKV in Mayotte in the discussion - please see lines 288-292.

I suggest the authors should assess for multicollinearity using VIF rather than just collinearity. 

Answer:

By using Spearman test we did consider multicollinearity rather than collinearity, as we checked the correlation byrunning the Spearman test with all the variables at once (although the output shows all possible combinations of correlation among each 2 variables). 

We did not opt for using the VIF command (https://www.ncbi.nlm.nih.gov/pmc/articles/PMC9901837/) as it is linked to the use of regression. We were using the poisson function, therefore we could not use the command VIF. The command collin could have been used instead as it does not require the regression function beforehand, but we opted for using the Spearman test as it seems to be more appropriate when variables are ordinal, not linear or not following a normal distribution, a situation that pretty much reflected our group of variables. 

I also wonder why the threshold of 0.5 was chosen for the pairwise correlation to reduce the number of variables. Why not 0.7 or any other numbers? 

Answer:

Based on some reading on Spearman test, the use of it, and the interpretation of thresholds - see for instance these: 

https://geographyfieldwork.com/SpearmansRankCalculator.html, 

https://statistics.laerd.com/stata-tutorials/spearmans-correlation-using-stata.php

https://www.ncbi.nlm.nih.gov/pmc/articles/PMC3576830/

we opted for choosing 0.5 as the threshold. The range between 0.5 and 0.7 is generally associated to moderate correlation. As we did not want to have correlation as a factor that would somehow impact the analysis, 0.5 seemed to be an appropriate and conservative threshold to use (alongside a significant P value).

I suggest the authors attempt to reduce the dimension of the variables by other means such as stepwise regression.

Answer:

We could not conduct an automated stepwise logistic regression using the stepwise command in STATA as the svy prefix command is not compatible with it. However, we did stepwise regression “manually”, specifically we used a backward stepwise process with the commands svy,:poisson and test.

On this note, the svy command was necessary as we could not avoid performing a weighed analysis in this case. https://www.stata.com/support/faqs/statistics/stepwise-regression-with-svy-commands/

The results from the univariate analysis should be summarized or discussed somewhere in the manuscript.

Answer:

We have taken on board your comment and briefly discussed the results from the univariate analysis. Please find the results in lines 241-250.

We opted for not adding any further point related to the univariable analysis in the discussion as not relevant to the most essential results of the study.

I also wonder why the authors perform the analyses on the three models with different fixed set of covariates. Some kind of variables selection procedure here may make the method more robust.

Answer:

After performing the Spearman test and eliminating (or simply recategorizing or combining) variables that were creating collinearity, we added all the remaining variables to one model. By doing the backward stepwise method and reaching a list of variables that were significative in the final model, we realized that we were possibly losing some relevant information that could have been useful to report. 

We then run several models with different set of variables, and eventually we reached the conclusion that based on the concept that a model should be parsimonious, it would have been preferable to use and report several models with a smaller number of variables than try to fit all the variables in one single model in which some of the information would have been lost.

Finally, we were not performing prediction models where it would be more appropriate to fit all the variables, but testing potential risk factors that may have been impacting the outcome we were measuring; therefore, (and considering the concept of model parsimony) it would have been preferable not to add all variables at once in the model.

“Based on an estimated basic reproduction number of 2.8 reported for areas where Aedes albopticus is the primary vector (29), roughly 65-70% of the population should be immune to CHIKV to avoid an epidemic; therefore, the possibility of future epidemics in Mayotte cannot be excluded.” I found this point to be problematic. The 65-70% figure is derived from the R0 of 2.8 assuming that the population is homogenous. This in reality is not true for aboviruses, where their transmission dynamic are highly heterogenous. Given the heterogeneity, the herd immunity threshold can be much lower than 65-70%, hence current immunity level in the population maybe enough to prevent epidemic to happen in the near future. Furthermore, the authors seems to ignore the uncertainty of the R0 estimation, which lead to the uncertainty in the herd immunity threshold as well. I also found the conclusion for this point is weak, since in the future, as the susceptible pool is increasing over time from the new-borns, it is inevitable that “the possibility of future epidemics in Mayotte cannot be excluded.”

Answer:

We do agree with your comment and we took off the reference to the Ro. We also added the % of the population under the age 15 that is susceptible and rephrased the paragraph. We also highlighted the resulting expected proportion of susceptible in the population of Mayotte. Please see lines 291-299.

“Knowledge of other mosquito-borne diseases such as malaria, appeared to be positively associated with the presence of CHIKV antibodies. This association could reflect a heightened awareness of mosquito-borne infectious diseases among individuals affected by malaria or chikungunya.” I failed to see the authors’ logic of this point. I believe there is a potential confounder here, knowledge of malaria and presence of CHIKV antibodies are both correlated with high risk of arboviruses infection.

Answer:

Our exposure of interest was “knowledge of other mosquito- borne diseases such as malaria” and the outcome is “CHIKV seropositivity”. This is the relationship we were looking at. A third variable that may have impacted this relationship is the strategies used to fight malaria between 2007 and 2014. During this period, people may have become more aware of malaria transmission and cases as prevention and surveillance for this disease were enhanced. (see this: https://malariajournal.biomedcentral.com/articles/10.1186/s12936-015-0837-6 )

If people had been already impacted by CHIKV infection during the 2005 outbreak, perhaps they may have been keener in understanding more about malaria as well, considering the similarities in transmission. We may infer that the exposure to CHIKV has possibly improved the knowledge of vector borne diseases in the population including malaria. We decided not to report this as it was as looking at the CHIKV as exposure and “knowledge of other mosquito- borne diseases such as malaria” as outcome.

The lines in the manuscript should be numbered.

Answer:

We did add the line number as suggested

Reviewer #2: 

Dear authors,

Excellent work conducting this work and conveying key information about a neglected disease. Please see some specific comments below:

Is it possible to add a map illustrating the different regions of Mayotte and their respective seroprevalence? A figure would be helpful for readers unfamiliar with the island when discussing the geographic findings of the study.

Answer:

Thank you for your suggestion, we have added a map with the requested information – see Fig 1.

Introduction, 2nd paragraph, line 4, remove “has” in “where presumably CHIKV has originated”

Introduction, final paragraph, line 4, change “insight of” to “insight into”

Setting, paragraph 1, line 3, change “is holding” to “holds”

Study design, paragraph 1, line 1, change “consists in” to “consists of”

Data collection, paragraph 2, line 7, change “Besides,” to “Additionally,”

Determinants of anti-CHIKV IgG seropositivity, paragraph 3, line 1, change “being resident either in Mamoudzou” to “residing in either Mamoudzou”

Determinants of anti-CHIKV IgG seropositivity, paragraph 3, line 2, remove “the” in “associated with the higher”

Determinants of anti-CHIKV IgG seropositivity, paragraph 3, line 3, change “being resident” to “residing”

Determinants of anti-CHIKV IgG seropositivity, paragraph 4, line 3, change “toilette” to “toilet”

Answer:

Thank you for highlighting all the lines where language changes were needed. We addressed them all.

Discussion, paragraph 2 - “therefore we should add this population to the group of susceptible individuals.” Is it possible to provide a quantitative estimate of this value? Ie. if you assume all members of the population below 15 are seronegative and calculate a new percentage including these individuals, by how much does the 65% susceptibility estimate increase?

Add % of the population 0-15 on top of 65%

Answer:

Thank you for this very useful suggestion. We calculated the % and added to the 65% - Please see line 291-296.

Reviewer #3: 

Ortu et al. performed a serosurvey and questionnaire in Mayotte, where a CHIKV outbreak took place in 2006. The authors performed IgG ELISA to determine previous CHIKV exposure and applied questionnaires exploring socio-demographic factors and knowledge of mosquito-borne diseases. Although the part of the study looking at CHIKV IgG could be substantially expanded upon, by doing quantitative ELISA, neutralization assays, and T-cell assays to assess the population immunity, the IgG data presented here will be a valuable addition to the literature for understanding CHIKV present and future outbreaks.

Major comments:

1.The title of the manuscript claims that the current immunity will not be enough to prevent future outbreaks. This claim is based on a report that estimates that at least 65% of the population would need to be immune to prevent future epidemics. However, in this paper the authors only perform a positive/negative IgG, and do not assess other parts of the immune system that could give partial immunity to CHIKV such as T-cells, and proportion of neutralizing antibodies, cross-neutralizing antibodies from similar alphaviruses. Therefore, the authors should consider changing the title to reflect their data accurately.

Answer:

Thank you for your comment – we do agree and we have modified the word in the title– we used “IgG seropositivity” rather than immunity 

2. The authors stored the sera at -20C. Please add how long the sera was stored for. How long are IgGs stable for at -20C?

Answer:

First, as appropriate for long-term stability (Hendriks et al. DOI: 10.4155/bio.14.96, Ma et al. https://doi.org/10.1016/j.biochi.2020.08.019), we stored samples at -20°C as we collected them between December 2018 and May 2019. Accounting for the collection period, storage, transportation delays due to COVID-19, a maximum of 2 years elapsed between collection and defrosting upon the analysis in the National Reference Centre for arboviral diseases in mainland France in Dec 2020. We added this information in the methods.

3. The references need to be improved on several occasions:

•The authors cite a manuscript in preparation; this is highly unusual, and I strongly suggest removing that citation. The details on the methodology should be included in this manuscript, in the methods or supplemental files, or a published source could be cited.

Answer:

Unfortunately, the publication of the cited source is still pending for lack of relevant reviewers; even after submission to PloS One, it had to be resubmitted elsewhere. To address this comment of Reviewer #3, we included a reference to the detailed methods as we did in a recently published study using the same samples (Bastard et al. doi: 10.1038/s43856-022-00230-4): Ruello, M. & Richard, J. Enquête de santé à Mayotte 2019—Unono Wa Maore. Méthode (Santé publique France, 2022, : https://www.santepubliquefrance.fr/docs/enquete-de-sante-a-mayotte-2019-unono-wa-maore.-methode ).

•References 15-17 look like they need more information, is that citing a book chapter?

Answer:

These are series that are published by Sante Publique France or regional public health agencies. In these cases, they are surveillance reports done by the agency in the Indian Ocean. We have amended the references and attached the reports.

•While citing reputable websites is acceptable, the authors cite many websites. I would suggest reducing the number of website citations wherever possible and adding the date accessed for any that will be kept. Links to websites tend to break with time and no longer be accessible.

Answer:

We have eliminated one web sites but left the others, which we thought were necessary. We added the access date and for a few we also included the pdf of the document. 

•Reference 31 is all in caps, while the others are not.

Answer:

We have amended it.

4. More details are needed regarding the ELISA. Please clarify if the ELISA is for total IgG, IgG-1 or other. Please specify how the ELISA threshold was chosen. If it was recommended by a manufacturer or based on previous experience or statistical method.

Answer:

For this serosurvey, we performed a homemade indirect ELISA, certified according to ISO 1589 standards. This assay using whole inactivated virus detected total IgG. More details and the following reference were added accordingly in manuscript (Denis J, et al. High specificity and sensitivity of Zika EDIII-based ELISA diagnosis highlighted by a large human reference panel. PLoS Negl. Trop. Dis. 2019;13:e0007747. doi: 10.1371/journal.pntd.0007747.)

5. Since the authors are working with human blood from a region that may have CHIKV present, the authors should specify how the samples were handled, ie/ level of containment, any heat inactivation done, if anything was done in case blood was collected from a patient with CHIKV symptoms. Although the samples were not known to contain CHIKV, this virus is typically BSL-3 in Europe and the US.

Answer:

Samples were opened and handled under a microbiological safety cabinet. Although participants were asymptomatic and their blood samples did not qualify as diagnostic specimens, they underwent a heat-inactivation step (56°C for 30 minutes).

Minor comments:

1. Would it be possible for the authors to use a known sample as standard curve to quantify the IgGs?

Answer:

Unfortunately, this does not apply because our ELISA method was qualitative.

2. The "attitudes to mosquito-borne disease prevention" data is not showing on table 2, I think it was cut off.

Answer:

There was actually one extra variable that was not supposed to be there – we have removed it. Thank you.

---

## [Decision Letter · Decision Letter 1]

11 Apr 2023

PONE-D-22-35076R1Long lasting anti-IgG chikungunya seropositivity in the Mayotte population will not be enough to prevent future outbreaks: A seroprevalence study, 2019.PLOS ONE

Dear Dr. Ortu,

Thank you for submitting your manuscript to PLOS ONE. After careful consideration, we feel that it has merit but does not fully meet PLOS ONE’s publication criteria as it currently stands. Therefore, we invite you to submit a revised version of the manuscript that addresses the points raised during the review process.

ACADEMIC EDITOR:Please address the remaining queries and suggestions by one of the reviewers. 

We look forward to receiving your revised manuscript.

Kind regards,

Joel Mossong, PhD

Academic Editor

PLOS ONE

Journal Requirements:

Reviewers' comments:

Reviewer's Responses to Questions

**Comments to the Author**

1. If the authors have adequately addressed your comments raised in a previous round of review and you feel that this manuscript is now acceptable for publication, you may indicate that here to bypass the “Comments to the Author” section, enter your conflict of interest statement in the “Confidential to Editor” section, and submit your "Accept" recommendation.

Reviewer #1: (No Response)

Reviewer #2: All comments have been addressed

2. Is the manuscript technically sound, and do the data support the conclusions?

Reviewer #1: Partly

Reviewer #2: Yes

3. Has the statistical analysis been performed appropriately and rigorously? 

Reviewer #1: Yes

Reviewer #2: Yes

4. Have the authors made all data underlying the findings in their manuscript fully available?

Reviewer #1: Yes

Reviewer #2: Yes

5. Is the manuscript presented in an intelligible fashion and written in standard English?

Reviewer #1: Yes

Reviewer #2: Yes

6. Review Comments to the Author

Reviewer #1: "The authors acknowledge your general feedback – thank you. We mentioned the evidence of not circulation of CHIKV in Mayotte in the discussion - please see lines 288-292."

No report does not mean no circulation of the virus. Please add the point I have made in my previous comment for the discussion. The positive proportion remains the same compared to the previous study in 2006 means that there are no or minimal transmission of the virus since then. If there was an outbreak recently, the positive proportion must be way higher than the previous one. This is a stronger argument for the no circulation in the region point and should be added to the discussion.

"By using Spearman test we did consider multicollinearity rather than collinearity, as we checked the correlation byrunning the Spearman test with all the variables at once (although the output shows all possible combinations of correlation among each 2 variables). We did not opt for using the VIF command"

No pair-wise co-linearity does not exclude the possibility of multicolinearity.

I do not get the difficulty to calculate VIF since the authors only need to calculate it based on the variables

"Based on some reading on Spearman test, the use of it, and the interpretation of thresholds - see for instance these:

https://geographyfieldwork.com/SpearmansRankCalculator.html,

https://statistics.laerd.com/stata-tutorials/spearmans-correlation-using-stata.php

https://www.ncbi.nlm.nih.gov/pmc/articles/PMC3576830/ we opted for choosing 0.5 as the threshold. The range between 0.5 and 0.7 is generally associated to moderate correlation. As we did not want to have correlation as a factor that would somehow impact the analysis, 0.5 seemed to be an appropriate and conservative threshold to use (alongside a significant P value)."

Please add these reference to the manuscript to explain why you choose the number.

"We could not conduct an automated stepwise logistic regression using the stepwise command in STATA as the svy prefix command is not compatible with it. However, we did stepwise regression “manually”, specifically we used a backward stepwise process with the commands svy,:poisson and test."

I am not familiar with STATA syntax so I cannot get what the authors trying to say here. Please use statistical terms to explain your reasoning.

Reviewer #2: Thank you for addressing the comments submitted by the reviewers. All of my comments and questions are now resolved.

7. PLOS authors have the option to publish the peer review history of their article (what does this mean?). If published, this will include your full peer review and any attached files.

Reviewer #1: No

Reviewer #2: No

---

## [Author Response · Author response to Decision Letter 1]

27 Apr 2023

Reviewer #1 

"The authors acknowledge your general feedback – thank you. We mentioned the evidence of not circulation of CHIKV in Mayotte in the discussion - please see lines 288-292."

No report does not mean no circulation of the virus. Please add the point I have made in my previous comment for the discussion. The positive proportion remains the same compared to the previous study in 2006 means that there are no or minimal transmission of the virus since then. If there was an outbreak recently, the positive proportion must be way higher than the previous one. This is a stronger argument for the no circulation in the region point and should be added to the discussion.

Answer:

We entirely concur. Therefore, we made our agreement clearer by adding the following (lines 287-288): “This estimate is very similar to the seroprevalence of anti-CHIKV IgG as found in 2006 just after the outbreak in Mayotte, suggesting that there was no or minimal transmission of the virus in the interim.”

"By using Spearman test we did consider multicollinearity rather than collinearity, as we checked the correlation by running the Spearman test with all the variables at once (although the output shows all possible combinations of correlation among each 2 variables). We did not opt for using the VIF command"

No pair-wise co-linearity does not exclude the possibility of multicollinearity.

I do not get the difficulty to calculate VIF since the authors only need to calculate it based on the variables

Answer:

As the reviewer requested, we calculated the VIF with all the variables that we wanted to use for the multivariable analysis (see page 3 of S6), and we obtained an average VIF of 1.62, with a variable showing a maximum of 3.2 VIF. Based on this average VIF value we would have used all the variables for the models, but we felt that this value was not enough to highlight the collinearity issues that we felt very likely present among some of these variables. Therefore, we run the Spearman test, that actually showed us which couples of variables were showing collinearity. Based on this test, we then decided to combine those variables that appear to be collinear, and/or eliminate one of the two that showed collinearity. The final list of variables that we then used for the running our models is the one showed in page 4 of the S6 information sheet. We run again the VIF test with these and the final average value was 1.52, a bit lower than before as we did address the suspected collinearity issues.

We hope that this explanation clarifies why eventually we decided initially, to report the Spearman test on our manuscript rather than the VIF, which we felt, did not provide a satisfactory answer to whether there was collinearity and for which variables. We have added now a few sentences to reflect the request of the reviewer and this issue. We hope that this amendment is acceptable for the reviewer (see lines : 188-190).

"Based on some reading on Spearman test, the use of it, and the interpretation of thresholds - see for instance these:

https://geographyfieldwork.com/SpearmansRankCalculator.html,

https://statistics.laerd.com/stata-tutorials/spearmans-correlation-using-stata.php

https://www.ncbi.nlm.nih.gov/pmc/articles/PMC3576830/ we opted for choosing 0.5 as the threshold. The range between 0.5 and 0.7 is generally associated to moderate correlation. As we did not want to have correlation as a factor that would somehow impact the analysis, 0.5 seemed to be an appropriate and conservative threshold to use (alongside a significant P value)."

Please add these references to the manuscript to explain why you choose the number.

Answer:

As initially we understood from the reviewers that it was preferred not to use many web sites among the references, we opted to add only the manuscript among the above references (see No: 26)

"We could not conduct an automated stepwise logistic regression using the stepwise command in STATA as the svy prefix command is not compatible with it. However, we did stepwise regression “manually”, specifically we used a backward stepwise process with the commands svy,:poisson and test."

I am not familiar with STATA syntax so I cannot get what the authors trying to say here. Please use statistical terms to explain your reasoning.

Answer:

By "manual stepwise regression ", we meant that we followed the alternate procedure presented on the Stata website: William Sribney. 2022. Is there a way in Stata to do stepwise regression with svy: logit or any of the svy commands? Stata FAQ. StataCorp Available at: https://www.stata.com/support/faqs/statistics/stepwise-regression-with-svy-commands/ Last Update Date: 16Nov2022. This website is referenced (see ref number: 27)

In brief, we arranged the selected covariates for each model into groupings ordered by decreasing presumed importance. Then we ran each full model before testing the last group of covariates. If not significant, the entire group was discarded. Otherwise, we kept the whole group. Then we repeated the procedure. Hopefully, adding it to our manuscript reference list will not pose a problem since we understood that the reviewers preferred that there were as few references to online resources as possible.

---

## [Decision Letter · Decision Letter 2]

4 May 2023

Long lasting anti-IgG chikungunya seropositivity in the Mayotte population will not be enough to prevent future outbreaks: A seroprevalence study, 2019.

PONE-D-22-35076R2

Dear Dr. Ortu,

We’re pleased to inform you that your manuscript has been judged scientifically suitable for publication and will be formally accepted for publication once it meets all outstanding technical requirements.

Kind regards,

Joel Mossong, PhD

Academic Editor

PLOS ONE

Additional Editor Comments (optional):

Reviewers' comments:

Reviewer's Responses to Questions

**Comments to the Author**

1. If the authors have adequately addressed your comments raised in a previous round of review and you feel that this manuscript is now acceptable for publication, you may indicate that here to bypass the “Comments to the Author” section, enter your conflict of interest statement in the “Confidential to Editor” section, and submit your "Accept" recommendation.

Reviewer #1: All comments have been addressed

2. Is the manuscript technically sound, and do the data support the conclusions?

Reviewer #1: Yes

3. Has the statistical analysis been performed appropriately and rigorously? 

Reviewer #1: Yes

4. Have the authors made all data underlying the findings in their manuscript fully available?

Reviewer #1: Yes

5. Is the manuscript presented in an intelligible fashion and written in standard English?

Reviewer #1: Yes

6. Review Comments to the Author

Reviewer #1: (No Response)

7. PLOS authors have the option to publish the peer review history of their article (what does this mean?). If published, this will include your full peer review and any attached files.

Reviewer #1: No

---

## [Editor Report · Acceptance letter]

9 May 2023

PONE-D-22-35076R2 

Long lasting anti-IgG chikungunya seropositivity in the Mayotte population will not be enough to prevent future outbreaks: A seroprevalence study, 2019. 

Dear Dr. Ortu:

I'm pleased to inform you that your manuscript has been deemed suitable for publication in PLOS ONE. Congratulations! Your manuscript is now with our production department. 

Kind regards, 

on behalf of

Dr. Joel Mossong 

Academic Editor

PLOS ONE